# Scalable inference of functional neural connectivity at submillisecond timescales

**Arina Medvedeva**
Flatiron Institute
New York, NY
amedvedeva@flatironinstitute.org

**Edoardo Balzani**
Flatiron Institute
New York, NY
ebalzani@flatironinstitute.org

**Alex H Williams**
Flatiron Institute, New York University
New York, NY
awilliams@flatironinstitute.org

**Stephen L Keeley**
Fordham University
New York, NY
skeeley1@fordham.edu

## Abstract

The Poisson Generalized Linear Model (GLM) is a foundational tool for analyzing neural spike train data. However, standard implementations rely on discretizing spike times into binned count data, limiting temporal resolution and scalability. Here, we develop Monte Carlo (MC) methods and polynomial approximations (PA) to the continuous-time analog of these models, and show them to be advantageous over their discrete-time counterparts. Further, we propose using a set of exponentially scaled Laguerre polynomials as an orthogonal temporal basis, which improves filter identification and yields closed-form integral solutions under the polynomial approximation. Applied to both synthetic and real spike-time data from rodent hippocampus, our methods demonstrate superior accuracy and scalability compared to traditional binned GLMs, enabling functional connectivity inference in large-scale neural recordings that are temporally precise on the order of synaptic dynamical timescales and in agreement with known anatomical properties of hippocampal subregions. We provide open-source implementations of both MC and PA estimators, optimized for GPU acceleration, to facilitate adoption in the neuroscience community[1].

## 1 Introduction

As recording technologies in neuroscience advance, there is a growing need to improve the scalability of statistical methods for analyzing neural spiking activity. A key challenge in understanding neural computation lies in accurately estimating functional connectivity—the statistical dependencies between neurons that reflect synaptic interactions. The Poisson Generalized Linear Model (GLM) is a powerful tool for this purpose, capable of inferring both stimulus encoding properties and coupling between spiking units. However, the standard implementation of the GLM requires binning the timeseries data into a large design matrix, $\mathbf{X}$, of discrete spike counts. The time resolution of this binning is often coarse ($\sim$ 1 to 10 ms) [1–5] compared to the timescale of synaptic dynamics, which rise and fall at submillisecond timescales [6–8]. This means conventional GLM implementations fail to capture synaptic coupling filters on a biophysically realistic scale [1, 3–5, 9]. Moreover, as the bin size decreases, $\mathbf{X}$ grows in size, posing significant computational and memory storage challenges.

---

[1]The Poisson point process GLM code is available at https://github.com/macari216/poisson-process-glm.git

39th Conference on Neural Information Processing Systems (NeurIPS 2025).

We find that for most modern neural datasets, storing $\mathbf{X}$ in memory is infeasible, requiring users to batch $\mathbf{X}$, which renders inference unstable even with state-of-the-art optimizers.[2]

Here, we propose methods that avoid these issues by considering the limit of infinitely small time bins, in which case the model becomes a Poisson point process (see e.g. Chapter 19 of [10]). Although point process models have been explored by the neuroscience community [11–17], most prior work either develops theoretical tools for continuous-time models without presenting fitting procedures (e.g., convexity of the log-likelihood [11] or error bounds [12]), or explores related model classes [15, 16], or uses numerical integration methods that do not scale to large datasets [17]; therefore, we limit our benchmark comparison to discrete-time GLM implementations [1, 4, 13, 14]. In our setting of interest, a point process model is able to capture fine-scale spike time correlations between co-recorded neurons, which can be indicative of monosynaptic connections [6, 7]. Furthermore, inputs to the model can be represented as a sequence of spike times instead of a large design matrix. However, to fit the point process model, we must numerically approximate an analytically intractable integral that appears in the likelihood function. We provide two approaches to deal with this integral: 1) a Monte Carlo sampling-based approach (MC) and 2) a second-order polynomial approximation, inspired by prior work [4, 18, 19]. Both methods demonstrate improvements in accuracy over conventional approaches while maintaining computational tractability. Additionally, the polynomial approximation yields a closed-form expression for the Poisson log-likelihood that is quadratic in the GLM parameters, enabling fast and efficient computation. We also propose generalized Laguerre polynomials scaled by an exponential as a new set of basis functions for GLM inference. While these polynomials retain the desirable temporal smoothing properties of the traditionally used raised cosine basis [20, 21], they offer orthogonality and closed-form integral solutions, enabling efficient filter identification.

We validate our models on both simulated and real spiking data. In simulations, we find that both MC and PA approaches scale favorably in compute time with recording length and population size, and show improved filter recovery compared to both the discrete polynomial approximate method and traditional GLMs. We then apply our method to real spiking data, where we analyze spike-time recordings from multiple rodent hippocampal regions [22] in a dataset whose size is computationally prohibitive for traditional batched GLMs. We show that recovered coupling filters align with empirical cross-correlograms (CCGs) with sub-millisecond temporal precision, suggesting the model is able to accurately identify monosynaptic coupling between neurons. In addition, we are able to use our model to isolate specific coupling filters that identify putative excitatory connections in the rodent hippocampus. We show that these isolated filters coincide with anatomical connectivity structure that is well-established in studies of hippocampal anatomy [23, 24], suggesting GLMs operating at this resolution provides new opportunities in the identification of neural circuitry from spike-train recordings.

## 2 Background

### 2.1 Discrete-time Poisson GLMs

Generalized linear models provide a useful tool for predicting spiking activity of a single neuron $\boldsymbol{y} = (y_1, \ldots, y_T)$ given recent population spiking activity or external stimuli $\boldsymbol{X} = (\boldsymbol{x}_1, \ldots, \boldsymbol{x}_T)$, and a set of model parameters $\boldsymbol{w}$. The spike counts $y_t$ are conditionally Poisson distributed, $y_t \sim \text{Poisson}(y_t|\boldsymbol{w}, \boldsymbol{x}_t)$, and the model log-likelihood is written as:

$$\log p(\boldsymbol{y} \mid \boldsymbol{X}, \boldsymbol{w}) = \sum_{t=1}^{T} y_t \log(\Phi(\boldsymbol{x}_t^T \boldsymbol{w})) - \Phi(\boldsymbol{x}_t^T \boldsymbol{w}) \tag{1}$$

where $\Phi(\boldsymbol{x}_t^T \boldsymbol{w})$ is the predicted firing rate at time bin $t$ and $\Phi : \mathbb{R} \to \mathbb{R}$ is a monotonically increasing, convex, and nonnegative function (e.g., exponential or softplus). The central goal of the Poisson GLM, in the identification of $\boldsymbol{w}$, is to find smooth time-varying statistical dependencies between either external stimuli or individual neuronal spike trains and post-synaptic firing rates in a neural population (Fig 1**A**). These filters are typically estimated using a linear combination of a small number

---

[2]While one can in principle represent $\mathbf{X}$ in a sparse matrix format to alleviate computational burden, there is currently limited support for sparse matrix routines in libraries that are compatible with modern GPUs.

of smooth basis functions and a nonlinearity to assure non-negative firing rates. The filters within neural populations reflect temporally delayed correlated firing, so called "functional connectivity," and are often thought of as a proxy to anatomical synaptic connections, reflecting how populations of neurons influence each other through either excitatory or inhibitory dynamics.

Throughout this work, we will focus primarily on estimating functional connectivity filters using the GLM, and we will use the exponential nonlinearity, $\Phi(\cdot) = \exp(\cdot)$, as this is a common choice in neuroscience and simplifies the log-likilhood objective. However, all of the methods here trivially work with an augmented $\mathbf{X}$ to include stimuli, and with alternative nonlinearities, such as softplus, which is another common choice in the field (see Supplement S.4 for more details).

The traditional approach described above requires discretization of the time series, with a bin size commonly chosen within the range from hundreds of milliseconds to one millisecond, depending on the system and stimulus (features) [1, 4, 25]. However, if the goal is to identify functional monosynaptic connections between neurons, which is a common motivation in modern GLMs, even 1 ms resolution is not sufficient. Electrophysiological recordings in experimental neuroscience have shown that synaptic dynamics are often highly transient, with the rise and fall in firing occurring within 1–5 ms following a presynaptic spike [26, 7]. This means that even bin sizes as small as 1 ms fail to accurately identify peak amplitude and timing (Fig 1**B** and **C**), which may be important for cell-specific synapse properties or distinguishing correlation firing patterns from synaptic activity.

For discrete-time GLMs, sampling at finer than 1 ms resolution demands prohibitively large memory allocations. The dimensionality of the feature space $\mathbb{R}^{NJ}$ depends on the number of neurons $N$ in the recording and the number of basis functions $J$ used to describe each neuron's activity history. For a given dataset, this results in a design matrix $\boldsymbol{X} \in \mathbb{R}^{T \times NJ}$. For long recordings from a large number of neurons, computing and storing this design matrix with a sufficiently small bin size becomes non-trivial. As shown in Fig 1**E**, simulating a dataset of 200 neurons at 1 ms or .1 ms resolutions for 10-100 minutes would require an $\mathbf{X}$ matrix of $10^{10}$–$10^{12}$ bits, necessitating batched gradient calculations. In contrast, storing only spike times drastically reduces memory usage, making GLM computations far more tractable for modern high-resolution (submillisecond) datasets.

While batching the design matrix $\mathbf{X}$ for discrete-time Poisson GLM optimization is a sensible approach, it poses significant problems when practically fitting the model. In particular, due to the sparse firing patterns of neural activity, the variance in gradients across batches can be very large. Even when implementing a state-of-the-art stochastic variance-reduced gradient (SVRG) optimization which guarantees an unbiased gradient estimates and minimal memory overhead [27], we find that in practice the variance of our updates is too large to achieve good fits as compared to discrete GLMs using small enough datasets to not require batching (Fig 3,4). Consequently, batched approaches are not only quite slow—requiring, for example, 5 hours on a dataset of 250 neurons with recording length 1000 seconds binned at 0.1 ms resolution—but they can lead to inaccurate model fits.

## 2.2   The Polynomial-Approximate GLM

Previous work has shown that approximating the nonlinearity in the Poisson likelihood with a polynomial can be effective tool for scaling GLMs [18, 19, 4]. These approaches use an orthonormal set of Chebyshev polynomials which provide a good approximation to GLM non-linearities over a wide range of values, and are effective even for just second order polynomial approximations [18]. Considering the exponential nonlinearity, the approximation can be written as $\exp(x)\Delta = a_2 x^2 + a_1 x + a_0$, where $\Delta$ is the time bin size and $a_2, a_1, a_0$ are the optimal Chebyshev coefficients that minimize the mean squared error between the nonlinearity and quadratic approximation across the specified range $[x_0, x_1]$. Using this approximation, the GLM log-likelihood can be written as:

$$\log p(\boldsymbol{y} \mid \boldsymbol{X}, \boldsymbol{w}) \approx \sum_{t=1}^{T} \boldsymbol{w}^\top \boldsymbol{x}_t^\top (y_t - a_1 \mathbf{1}) - a_2 \boldsymbol{w}^\top \boldsymbol{x}_t^\top \boldsymbol{x}_t \boldsymbol{w} \tag{2}$$

where terms that do not depend on $\mathbf{w}$ are dropped, and $\mathbf{1}$ is a vector of ones. Because the log-likelihood is quadratic in the parameters, one can directly compute a maximum a posteriori (MAP) estimate using the sufficient statistics ($\sum_{t=1}^{T} \boldsymbol{x}_t$, $\sum_{t=1}^{T} y_t \boldsymbol{x}_t$, and $\sum_{t=1}^{T} \boldsymbol{x}_t \boldsymbol{x}_t^\top$). For more information on this approach, see [4].

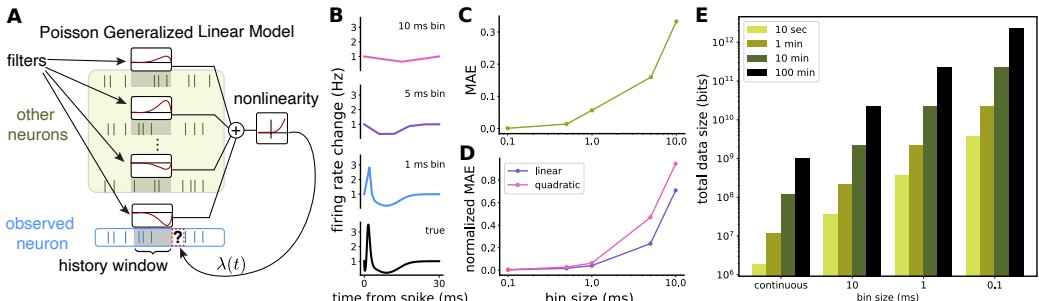

Figure 1: **A** Schematic of GLM for neuronal filter identification; **B** Simulation of realistic timescale post-synaptic conductance change as estimated by a GLM binned at 10, 5 and 1 ms bins; **C** Mean absolute error (MAE) on filter accuracy from **B** at various bin sizes; **D** Normalized error of discrete-time sufficient statistics from continuously generated Poisson rates estimated using various bin sizes; **E** Memory storage of spike times and **X** for 200 neurons at various recording lengths and bin sizes.

We find that the second-order polynomial approximation is helpful in significantly reducing the computational time of the GLM, but batched sufficient statistics can still carry a large computational load and can be time-consuming on datasets with fine temporal resolution. Moreover, the binning of the design matrix introduces an error in the estimation of the linear and quadratic sufficient statistics that accumulates with increasing number of spikes in the recording (Fig 1**D**) (see Supplement S.6.2 for more details).

## 3 The Poisson process GLM model

To improve the scalability and accuracy of these traditional GLM approaches, we instead consider a continuous-time Poisson Process GLM log-likelihood given by:

$$\log p(\boldsymbol{y} \mid \boldsymbol{X}, \boldsymbol{w}) = \sum_{k=1}^{K} \log \lambda(y_k) - \int_0^T \lambda(t)\, dt \tag{3}$$

Here, a time-varying Poisson rate $\lambda(t)$ is evaluated at time points designated by observed spike times $y_k$ of the post-synaptic neuron $\boldsymbol{y} = (y_1, \ldots, y_K)$, and the second term integrates the rate over the duration of the entire recording $[0, T]$. The firing rate at time $t$ is then given by:

$$\lambda(t; \boldsymbol{X}, \boldsymbol{w}) = \Phi\left[\sum_{\boldsymbol{x}_s \in \mathcal{X}(t, H)} \boldsymbol{w}_{n_s}^\top \boldsymbol{\phi}(t - t_s)\right] \tag{4}$$

where $\mathbf{X} = (\boldsymbol{x}_1, \ldots, \boldsymbol{x}_S)$ represents the full set of $S$ spikes and each spike $\boldsymbol{x}_s = (n_s, t_s)$ indicates that neuron $n_s \in 1, \ldots, N$ fired at time $t_s$; $\mathcal{X}(t, H)$ denotes the set of spikes occurring in the history window $[t - H, t]$; $\boldsymbol{w}_{n_s} \in \mathbb{R}^J$ is a subset of weights associated with neuron $n_s$; and $\boldsymbol{\phi} : [0, H] \to \mathbb{R}^J$ denotes a nonlinear mapping onto $J$ temporal basis functions. In this work, we select history window length $H$ of 4-6 ms to encompass expected neuronal dynamical effects. While $\mathbf{X}$ can be easily augmented to include external stimuli, here we restrict our analysis to spike history, primarily focusing on the role of neural interactions and intrinsic dynamics at synaptically relevant timescales.

Given that the intensity function $\lambda(t)$ is defined analytically, the first term in the Poisson process log-likelihood can be computed exactly. However, the nonlinearity $\Phi$ makes the cumulative intensity function (CIF) $\int_0^t \lambda(\tau)\, d\tau$ intractable, and thus the second term of the log-likelihood requires approximation. Here, we propose two methods to approximate this integral: 1) a Monte Carlo sampling-based approach (MC) with an unbiased estimator for the CIF; and 2) a polynomial approximation (PA) that yields an expression quadratic in the GLM parameters.

### 3.1 Monte-Carlo sampling for the CIF

To compute the second term in the objective function, $\int_0^T \lambda(t)dt$, we approximate the integral with a Monte Carlo estimate. Instead of simple uniform sampling, we employ stratified sampling: the time support $[0, T]$ is divided into $M$ equal subintervals, and sample points $\boldsymbol{\tau} = (\tau_1, \ldots, \tau_M)$ are drawn uniformly from each subinterval. Then,

$$\frac{T}{M} \sum_{m=1}^M \lambda(\tau_m) \approx \int_0^T \lambda(t)dt \tag{5}$$

provides an unbiased estimator of the integral that exhibits lower variance compared to uniform Monte Carlo sampling (see Chapter 8 in [28]). Thus, our loss function for a fixed sample of $\boldsymbol{\tau}$ is:

$$f(\boldsymbol{w}, \boldsymbol{\tau}) = \frac{T}{M} \sum_{m=1}^M \lambda(\tau_m) - \sum_{k=1}^K \log \lambda(y_k) \tag{6}$$

Where the second term can be computed exactly. We can employ standard gradient-based optimization procedures on this objective selecting a different $\boldsymbol{\tau}$ at every iteration.

### 3.2 The Polynomial-Approximate continuous GLM

Alternatively, we can use a polynomial approximation method inspired by Zoltowski and Pillow [4] and Huggins et al. [18] to derive a tractable, scalable form for the log-likelihood's CIF. By fitting a second-order polynomial with coefficients $a_2, a_1, a_0$ to minimize the mean squared error (MSE) against the true nonlinearity over a specified range, we reformulate the objective into a sum of integrals over linear terms (individual basis functions) and quadratic terms (basis function pairs). Depending on the choice of basis functions, these integrals may admit analytic solutions, enabling efficient evaluation of the log-likelihood. The polynomial-approximate CIF is written as:

$$\begin{aligned}
\int_0^T \lambda(t)dt &= \int_0^T \Phi\left( \sum_n \sum_{t_s \in \mathcal{X}_n} \boldsymbol{w}_n^\top \boldsymbol{\phi}(t - t_s) \right) dt \\
&\approx a_2 \int_0^T \left( \sum_n \sum_{t_s \in \mathcal{X}_n} \boldsymbol{w}_n^\top \boldsymbol{\phi}(t - t_s) \right)^2 dt + a_1 \int_0^T \sum_n \sum_{t_s \in \mathcal{X}_n} \boldsymbol{w}_n^\top \boldsymbol{\phi}(t - t_s)\, dt + T a_0 \\
&= a_2 \boldsymbol{w}^\top \mathbf{M} \boldsymbol{w} + a_1 \mathbf{m}^\top \boldsymbol{w} + T a_0
\end{aligned} \tag{7}$$

Here, $\mathcal{X}_n$ denotes the set of spikes from neuron $n$ and the linear term includes a defined vector $\mathbf{m} \in \mathbb{R}^{NJ}$ that contains $N$ concatenated $\boldsymbol{\varphi}$ vectors scaled by respective total number of spikes per neuron, $S_n$: $(\mathbf{m} = S_1 \boldsymbol{\varphi}, S_2 \boldsymbol{\varphi} \ldots S_N \boldsymbol{\varphi})$, where $\boldsymbol{\varphi}$ is a vector of precomputed integrals for each of the $J$ basis function over $\tau = t - t_s$. That is, $\varphi_j = \int_0^H \phi_j(\tau)d\tau$.

The quadratic term is a symmetric block matrix $\mathbf{M} \in \mathbb{R}^{NJ \times NJ}$ with $N \times N$ blocks of size $J \times J$. Each block $\mathbf{M}_{n,n'}$ corresponds to a neuron pair $(n, n')$ and accumulates the contributions from all spike pairs $(t_s, t_{s'})$ with $t_s \in \mathcal{X}_n$ and $t_{s'} \in \mathcal{X}_{n'}$. The entry at position $(j, j')$ of the block is given by:

$$[\mathbf{M}_{n,n'}]_{j,j'} = \sum_{\substack{t_s \in \mathcal{X}_n \\ t_{s'} \in \mathcal{X}_{n'}}} \int_{\delta_{t_s,t_{s'}}}^H \phi_j(\tau)\phi_{j'}(\tau - \delta_{t_s,t_{s'}})\, d\tau, \tag{8}$$

where $\delta_{t_s,t_{s'}} = |t_s - t_{s'}|$ is the spike time difference. This integral is nonzero only when $\delta_{t_s,t_{s'}} \leq H$, i.e., when the spike pair is within the interaction window. Therefore, if these basis function products can be expressed analytically and integrated in closed form, we only need to compute all pairwise spike time differences within the window $[t_s - H, t_s]$ and sum the $J \times J$ integral evaluations.

Given the quadratic expression of the CIF, the first term of the log-likelihood can be computed exactly when using the exponential inverse link function. The contributions from presynaptic spikes are precomputed as neuron-specific vectors $\boldsymbol{\psi}_n = \sum_{k=1}^K \sum_{t_s \in \mathcal{X}_n(y_k, H)} \boldsymbol{\phi}(y_k - t_s)$, yielding the compact

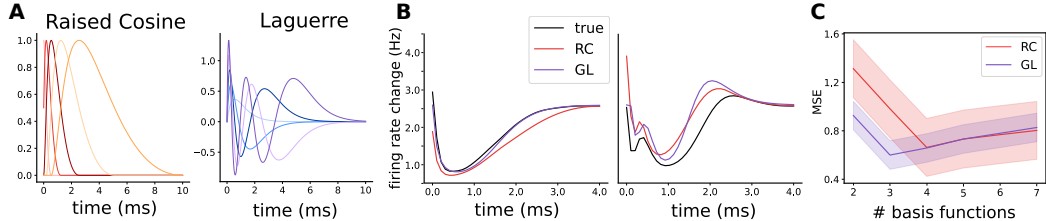

Figure 2: **A** Visualization of the first 5 RC and GL basis functions; **B** Best-performing fits of both bases onto filters generated from 100 RC bases; **C** Error on filter reconstruction for varying number of bases for both models.

form $\sum_{n=1}^{N} \boldsymbol{w}_n^\top \boldsymbol{\psi}_n = \boldsymbol{w}^\top \mathbf{k}$ where $\mathbf{k} \in \mathbb{R}^{NJ}$ concatenates all $\boldsymbol{\psi}_n$. Now, the full log-likelihood can be approximated as:

$$
\begin{aligned}
\log p(\mathbf{y} \mid \mathbf{X}, \boldsymbol{w}) &= \sum_{k=1}^{K} \log \lambda(y_k) - \int_0^T \lambda(t)dt \\
&\approx \boldsymbol{w}^\top (\mathbf{k} - a_1 \mathbf{m}) - a_2 \boldsymbol{w}^\top \mathbf{M} \boldsymbol{w}
\end{aligned}
\tag{9}
$$

which admits a closed-form solution for model parameters $\boldsymbol{w}$. For additional details, the full derivation of the quadratic polynomial approximation to the Poisson process log-likelihood and its extension to non-canonical link functions (e.g., softplus), please refer to Supplement S.4 and S.6.

We define the approximation range for the nonlinearity $\Phi$ based on estimates of the postsynaptic neuron's firing rate. In simulations, where ground-truth binned firing rates are available, the approximation range is set between the 2.5th and 97.5th percentiles of these rates, mapped back through the inverse link function (i.e., $\log(\cdot)$ when $\Phi = \exp$). For real data, where firing rate distributions are not directly accessible, we center the range at the inverse of the mean firing rate and determine its bounds by maximizing cross-validated log-likelihood, following the approach of [4]. In our analyses of neural recordings, we use an approximation interval spanning 3–7 Hz around the mean rate. Notably, wider intervals accommodate more variability in the estimated filter amplitudes but increase approximation error. As a result, polynomial approximation methods produce higher error when estimating the true underlying filters (simulated data) or CCGs (real data, see Figs. 4**B**, **D**, and 5**B,C**).

### 3.3 Generalized Laguerre polynomials as basis functions

We propose using scaled generalized Laguerre (GL) polynomials as basis functions for GLM temporal filters. Unlike raised-cosine (RC) bases, these functions are orthogonal under the weight $t^\alpha e^{-t}$ and thus can provide more efficient representation of filter variability with fewer basis functions [29]. These polynomials have the added feature of following an approximate gamma-function envelope, in line with fine time-scale rises and slow decays that correspond to biophysical synaptic and neuronal dynamics (Fig 2**A**). The parameter $\alpha > -1$ controls the long time-scale delay of the filter, $\alpha = 0$ yielding standard Laguerre polynomials. We additionally add a coefficient $c$ to the input variable $t$ that scales the rise-time of the bases. We set $c = 1.5$ and $\alpha = 2$ throughout the manuscript based on initial model exploration, but find that varying these values does not dramatically change model performance (Fig. S2**D**).

These orthogonal polynomials better capture filters in fewer basis functions than the standard RC basis. We demonstrate this on a simulated all-to-one coupled GLM whose filters are generated from 100 raised cosine bases. We simulate an 8-neuron population over a 1000-second recording, with the postsynaptic neuron's baseline firing rate set to 3 Hz. On these data, we fit the continuous MC GLM using either the standard RC or GL sets of 2-7 bases. We find coupling filters are better matched using GL in fewer bases functions, with the best performing model being 3 GL bases. Figure 2**B** shows filter matches using 3 GL and 4 RC bases, and 2**C** shows the mean error $\pm$ standard deviation across all simulated filters. For more details on the properties of the generalized Laguerre basis and comparison to RC, refer to Supplement S.5.

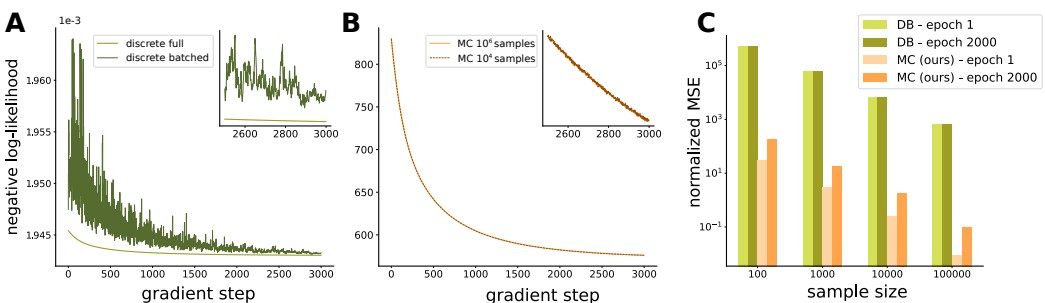

Figure 3: **A** First 3000 evaluations of negative log-likelihood objective on simulated data for batched and full discrete GLM with batch size = $10^4$ time bins; **B** Same as **A** for the continuous GLM with MC optimizer for sample sizes comparable to the evaluations in **A**; **C** Normalized MSE of the stochastic gradients relative to the full gradient at the beginning and end of the optimization procedure for discrete and MC models, across different batch and sample sizes.

These bases also have the advantage of admitting straightforward closed-form solutions for both single and pairwise product basis function integrals. Given a generalized Laguerre polynomial of degree $n$ and parameter $\alpha$, noted by $L_n^{(\alpha)}(ct)$, integrals $I_n$ of these bases have the form:

$$I_n = \int_0^H L_n^{(\alpha)}(ct) ct^{\alpha/2} e^{-ct/2} \, dt = \sum_{k=0}^n C_n \int_0^H t^{k+\alpha/2} e^{-ct/2} \, dt \tag{10}$$

where $C_n$ is a polynomial constant that depends on $n$ and $\alpha$. This admits exact integration via the lower incomplete gamma function $\gamma(a, x) = \int_0^x t^{a-1} e^{-t} \, dt$, with similar closed-form solutions available for pairwise basis function evaluations (see Supplement S.5 for derivations). While our polynomial approximation framework does not strictly require analytical solutions—as numerical integration remains computationally efficient—we found that using these closed-form expressions yielded optimal performance in both accuracy and speed for our implementation. For the remainder of this work, we run all simulations with 100 RC bases and fit all models with 3 to 5 GL bases.

## 4    Experiments

### 4.1    Stochastic gradient variance in discrete and continuous GLM

We first show that a naive approach to implementing traditional GLMs on modern datasets—batching the design matrix $\mathbf{X}$—fails to converge to the optimum due to high variance of gradient estimates across batches. The discrete batched (DB) approach performs parameter updates on small subsets of data, resulting in highly inaccurate gradients. When comparing DB to the full approach (on datasets small enough for the full design matrix $\mathbf{X}$ to fit in memory), we find that the GLM log-likelihood converges poorly under gradient descent in the batched case, failing to reach the global optimum achieved by the unbatched version (Fig. 3**A**). This gradient variability is a function of batch size, but even for batch sizes that push memory limits, gradient error remains prohibitively high on large datasets (Fig 3**C**). We therefore look to other approaches for scaling GLMs to large datasets.

Our Monte Carlo (MC) approach also introduces stochasticity in gradient estimates as different samples approximate the CIF integral. However, this variability is substantially lower than that of the discrete batched approach, resulting in much more stable inference with better log-likelihood values (Fig. 3**B**). This improved stability arises from two key differences: first, the spike term (first term in the log-likelihood) is always computed exactly over all observed spikes rather than a subset; second, although MC sample size affects the accuracy of the CIF integral estimate (the second term), stratified sampling ensures uniform coverage of the entire recording duration.  In Fig. 3**C**, we quantify the resulting improvement in gradient accuracy by computing the expected squared error between the true and stochastic gradients, normalized by the squared norm of the initial gradient and averaged over all parameter components:

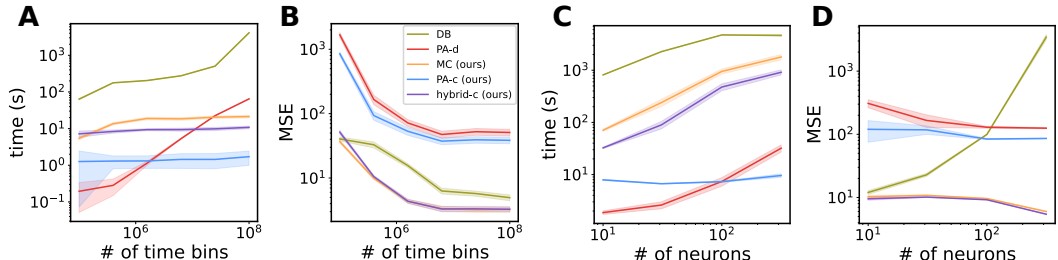

Figure 4: **A** Time to completion for discrete batched GLM (DB), discrete PA (PA-d), and our continuous models as recording length increases in size; **B**, filter accuracy for the models in **A**; **C**, same as **A** as the number of neurons in the population increases; **D**, same as **B** for the model fits in **C**.

$$\nu_p = \mathbb{E} \left[ \frac{||\nabla_p - \tilde{\nabla}_p||_2^2}{||\nabla_1||_2^2} \right] \tag{11}$$

where $\nabla_p$ is the true gradient at step $p$ and $\tilde{\nabla}_p$ is the corresponding stochastic gradient. Throughout inference, this error remains orders of magnitude higher in the DB model compared to the continuous sampling-based MC approach. Note that the error increases toward the end of training for both methods, as accurately estimating increasingly small gradient steps becomes more difficult as models approach convergence.

## 4.2 Continuous GLM scalability

We compare model performance and runtime across five approaches: a DB GLM with an SVRG optimizer [27] (DB); the polynomial approximation method of Zoltowski and Pillow [4] (PA-d); our continuous-time polynomial approximation (PA-c); our sampling-based Monte Carlo method (MC); and a hybrid approach that initializes MC inference with PA-c estimates (a "warm start"), reducing optimization steps and accelerating convergence. First, we evaluate performance on simulated data from an all-to-one coupled GLM ($N = 8$), varying recording duration from 10 to $10^4$ seconds, which spans the range of modern neuroscience recordings, with the bin size set to 0.1 ms for discrete models. (Fig. 4**A**,**B**). Next, we assess scalability by simulating a random, sparsely (10%) connected GLM with increasing population size ($N = 10$ to $N = 350$) with a fixed recording length $T = 100$ sec (Fig. 4**C**). We evaluate model performance by computing the mean squared error (MSE) between the estimated and true filters.

While SVRG guarantees convergence given enough passes through the full data, in practice we find that even when its runtime exceeds that of all other models by orders of magnitude, DB still underperforms, which is particularly evident at larger population sizes (Fig. 4**D**). The PA-d method is computationally efficient for smaller dataset sizes but eventually scales poorly in time and neuron number due to the cost of batch-computing sufficient statistics. In contrast, continuous-time methods utilize GPU-parallelized scans over the data, making them largely insensitive to recording length while increase only moderately with population size (Fig. 4**A**,**C**). In terms of estimation accuracy, the polynomial approximation methods (PA-d and PA-c) are less accurate, as expected, due to their approximations in the log-likelihood. The MC and hybrid models achieve the best filter recovery, with the hybrid approach offering the best tradeoff between speed and accuracy (Fig. 4**B**,**C**). We note here also that PA-c slightly outperforms PA-d due to inaccuracies present in binned data, though both use identical nonlinearity and approximation ranges. Further discussion of the discretization error and example filters from all models are provided in Supplement S.2.

## 4.3 Evaluation on hippocampal data

The hippocampus is a highly interconnected brain region essential for memory formation and retrieval. Its canonical trisynaptic circuit comprises the dentate gyrus (DG), CA3, and CA1 subregions, with distinct connectivity: the DG projects sparsely to CA3 via mossy fibers, with reciprocal connections back from CA3, and CA3 drives CA1 via the Schaffer collaterals (Fig.5**A**). Additionally, CA3 exhibits

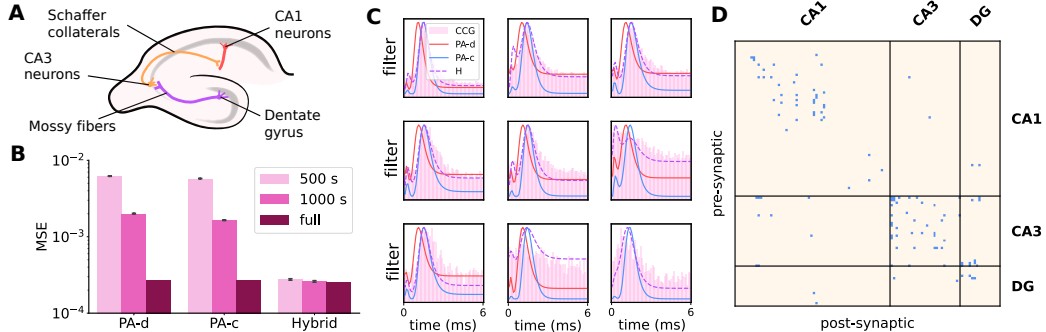

Figure 5: **A** Schematic of hippocampal anatomy; **B** Alignment of filter estimates on subsets of data with CCGs calculated from full dataset; **C** Example estimated filters with overlaid CCGs selected from high firing rate neuron pairs; **D** Putative excitatory connections across hippocampal subregions.

dense recurrent excitatory (EE) connectivity—a hallmark feature supporting autoassociative memory dynamics [23]. While this anatomical framework is well-established [24, 30], inferring monosynaptic connectivity and population-level spiking dynamics from multi-region electrophysiological recordings remains a significant statistical challenge. Cross-correlograms (CCG) based methods are computationally demanding at large scale and require additional processing to extract interpretable synaptic coupling patterns [31, 32, 25]. In contrast, GLMs offer a compact, efficient alternative that reduces parameter count while capturing temporal structure. This setting thus presents an opportunity to evaluate our continuous-time GLM models, which operate at submillisecond temporal resolution.

We use publicly available data from the Allen Institute consisting of 106 neurons ($N_{CA1} = 62$, $N_{CA3} = 28$, $N_{DG} = 16$) recorded with a single probe over approximately 2.7 hours [22]. All models are run with ridge regularization ($\beta = 1000$), a common choice for GLMs [3, 4, 9], to encourage sparsity in synaptic connections (see Supplement S.1 for more hyperparameter details). To assess filter accuracy, we compute the MSE between CCGs calculated on the full dataset and filter estimates from hybrid PA-MC (H), PA-c, and PA-d models on various subsets of the data. We exclude the discrete batched model (DB) from this analysis, as running it to convergence on the full dataset would be computationally infeasible. We find that our filters empirically match the pairwise CCGs, with the hybrid model showing the closest alignment even with only 500 seconds (8.3 minutes) of data, a small fraction of the full 2.7-hour recording (Fig.5**B, C**). While CCGs serve as a proxy for putative connections and cannot fully isolate synaptic effects from common input or indirect pathways, they provide a useful benchmark for evaluating filter estimates. Furthermore, after pre-selecting filters with peaks between 0.3–2.5 ms—indicative of excitatory connections—we find a connectivity structure that closely reflects known hippocampal anatomy (Fig. 5**D**, Table 1). The CA3 network exhibits the highest density of recurrent excitatory connections ($\sim 4\%$), consistent with anatomical estimates [24], while also showing bidirectional communication with the dentate gyrus [30] and Schaffer collateral projections to CA1 (Table 1. Notably, cross-region couplings tend to exhibit longer temporal delays (measured as time from filter onset to peak) than intra-regional latencies, consistent with axonal conduction times between structures and suggesting physiological validation of our identified connections. Fit results on the full dataset ($N = 623$ neurons across all probes) and a comparison showing improved performance with Generalized Laguerre versus raised cosine basis functions are provided in Supplement S.1.3.

## 5   Conclusion

We developed a continuous-time GLM implementation capable of identifying fine-timescale coupling filters in modern large-scale neural recordings, rendering modern datasets (hundreds of neurons recorded for thousands of seconds) trainable in minutes with sub-millisecond precision. Our focus has been on detecting potential synaptic connections through coupling filters, complementing existing approaches [7, 31], with a key advantage of being able to rapidly screen candidate connections in large datasets.

Table 1: Putative excitatory connections and synaptic latencies across hippocampal regions.

| Block | Pairs Total | Putative E | Fraction (%) | Mean Delay (ms) |
|---|---|---|---|---|
| CA3→CA3 | 784 | 30 | 3.83 | $1.75 \pm 0.52$ |
| DG→DG | 256 | 9 | 3.52 | $0.85 \pm 0.31$ |
| CA3→DG | 448 | 12 | 2.68 | $1.57 \pm 0.83$ |
| CA1→CA1 | 3844 | 51 | 1.33 | $1.69 \pm 0.75$ |
| DG→CA3 | 448 | 5 | 1.12 | $2.09 \pm 0.33$ |
| CA3→CA1 | 1736 | 18 | 1.04 | $2.15 \pm 0.34$ |
| CA1→DG | 992 | 4 | 0.4 | $1.86 \pm 0.29$ |
| CA1→CA3 | 1736 | 3 | 0.17 | $2.17 \pm 0.06$ |

Our work complements existing continuous-time modeling efforts which have different modeling goals or operate in smaller data regimes. In particular, Hawkes processes [16] represent a computationally efficient approach to identifying excitatory neuronal connections, but they cannot model inhibitory connections and thus occupy a different model class than the general Poisson process GLM. Other models, such as continuous Point-process latent variable models [33], share a similar likelihood construction but focus on identifying latent structure rather than fine-scale functional connectivity. To our knowledge, the only prior work that actually fits a continuous-time GLM [17] uses Gauss-Lobatto quadrature to approximate the integral in the log-likelihood. However, this approach requires inserting quadrature nodes between every spike time, making it computationally infeasible for the dataset sizes explored here (see Supplement S.3 for details). These fundamental limitations—model structure mismatch and computational infeasibility—precluded direct comparison to these methods in our benchmarks.

Our approach inherits several limitations from the broader class of Poisson GLMs, including the challenge of dissociating monosynaptic connections from correlated firing [2] and the difficulty of identifying true connectivity without overly penalizing weak dependencies or connections involving low-firing neurons [25, 32]. The Poisson distribution itself may be suboptimal for describing neural spiking due to its variance assumptions; flexible alternatives such as the negative binomial distribution [19, 34] could better capture spiking characteristics. Additionally, there is a fundamental trade-off between our two approximation methods: the PA approach enables faster inference through closed-form solutions but is inherently less accurate due to its global approximation of firing rates, while the MC approach is more accurate but requires multiple iterations to converge. Our hybrid model, which uses PA-based initialization followed by MC finetuning, is our attempt to balance this trade-off.

Key future directions include: more thorough evaluation of sparsity priors for population recordings, use of additional non-linearities, per-neuron approximation range optimization for our polynomial-approximate approach, and exploring variance reduction techniques [17, 35] for Monte Carlo sampling of the CIF. Additionally, extending the framework to incorporate latent population dynamics—for instance, by modeling shared low-dimensional trajectories at slower timescales similar to GPFA [33]—could help disentangle fast coupling dynamics from slower coordinated population activity, potentially improving both interpretability and generalization to held-out neurons.

# 6   Acknowledgments

This work was supported by the Simons Foundation. AHW was supported by the NIH BRAIN initiative (1RF1MH133778).

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
