# OpenReview forum: "Scalable inference of functional neural connectivity at submillisecond timescales"
_NeurIPS.cc/2025/Conference — NeurIPS 2025 poster_

### Official Review · Reviewer_8hfS · 2025-06-22

**Clarity:** 3
**Significance:** 3
**Originality:** 3
**Rating:** 4
**Confidence:** 4

**Summary:**

This paper presents novel scalable inference methods for functional neural connectivity using continuous-time Poisson Generalized Linear Models (GLMs), directly addressing the limitations of traditional binned GLMs. Conventional GLMs struggle to capture submillisecond synaptic dynamics due to coarse binning and become computationally prohibitive for large-scale recordings.

The authors introduce two primary approaches that operate in continuous time: a Monte Carlo (MC) sampling-based method and a polynomial approximation (PA) method. Both methods demonstrate improved accuracy and computational tractability by avoiding the need for a large design matrix. A significant contribution is the proposal of exponentially scaled Laguerre polynomials as orthogonal temporal basis functions, which enhance filter identification and yield closed-form integral solutions for the PA approach.

Validated on synthetic and real rodent hippocampal data, these continuous-time methods achieve high accuracy and scalability, enabling functional connectivity inference with submillisecond temporal precision. The recovered coupling filters align well with empirical cross-correlograms and established anatomical properties of hippocampal subregions, providing new opportunities for neural circuitry identification.

**Questions:**

- It would be helpful to include more comparisons or discussion with other continuous-time GLM approaches.
- Maximum likelihood estimation in autoregressive GLMs is often prone to instability issues, such as self-excitation. Does the proposed method encounter similar challenges?
- How does the overall firing rate influence the performance of the method?

**Ethical Concerns:**

["NO or VERY MINOR ethics concerns only"]

**Final Justification:**

I am mostly satisfied with the response, and will maintain my score.
The paper successfully showcases high accuracy and temporal precision, efficient basis functions, and tackles the scalability challenge for modern large-scale neural recordings. Reproducibility and detailed experimental setup are well-covered.
The remaining concern is the limited comparison or discussion with other existing continuous-time GLM approaches. A more explicit contextualization would be valuable. The authors acknowledged self-excitation instability in autoregressive GLMs applies, though overall stability was observed in simulations.

**Limitations:**

Yes

**Quality:**

3

**Strengths And Weaknesses:**

Strengths
- The work directly tackles the issues of coarse temporal resolution and scalability inherent in traditional binned Poisson GLMs, which struggle to capture submillisecond synaptic dynamics and become computationally prohibitive for large-scale recordings due to the need for a large design matrix.
- High accuracy and temporal precision: The continuous-time approaches demonstrate superior accuracy and scalability in filter recovery, enabling accurate identification of monosynaptic coupling.
- Efficient basis functions: exponentially scaled Laguerre polynomials as orthogonal temporal basis functions, which improve filter identification, provide closed-form integral solutions for the PA approach, and better capture biophysical synaptic dynamics compared to raised-cosine bases.

Weaknesses
-   The polynomial approximation methods (both PA-d and PA-c) are inherently less accurate compared to Monte Carlo (MC) or hybrid approaches.
-   While the Monte Carlo (MC) approach improves stability compared to discrete batched GLMs, the paper identifies "exploring variance reduction techniques for Monte Carlo sampling of the CIF" as a key future direction.
-   The current work primarily focuses on identifying functional connectivity using an exponential nonlinearity and restricts its analysis to spike history.
-   Related work: existing work on continuous-time GLM e.g. https://doi.org/10.1109/IEMBS.2009.5334610 is not discussed in the paper.

---

> ### Author Rebuttal · Authors · 2025-07-31
>
> We thank the reviewer for their time and effort reviewing our submission, as well as their helpful comments.
>
> ### **Global Response**
>
> **Comparison to existing literature/baseline evaluation**
>
> All reviewers note that there is prior work on continuous GLMs and note a lack of comparison to these methods. However, much of the cited work are not actual model fits of a continuous GLM, and thus are not appropriate baselines for direct comparison. Some citations develop theoretical tools relevant to continuous Poisson point processes—such as convexity of the log-likelihood [1], or error bounds and information capacity [14]—and others use continuous-time models in latent variable settings [16]. Some of the other work mentioned, particularly [15] and [Chen2019] referenced by 8hfS are actually discrete-time implementations of GLMs, and so are captured in our discrete benchmark in figures 3, 4, and 5.
>
> To our knowledge, the only prior work that actually fits a continuous time Poisson GLM is [17] which uses Gauss-Lobatto quadrature to approximate the CIF. Unfortunately, the link provided in their manuscript to public code is no longer available. We attempted to implement their method but found that a naive implementation was both computationally infeasible on datasets of the size we consider and less accurate in approximating the CIF and fitting the model on very small subsets of data. Their approach relies on inserting a varying number of quadrature nodes between every spike time, which, for our hippocampal dataset with ~5 million spikes, would require storing and evaluating far more nodes than spikes—leading to significant memory issues and prohibitively slow inference (MC, by contrast, uses many times fewer samples than spikes). We have since tested their quadrature approach on dataset smaller than our current evaluations in the manuscript and found that the CIF estimate as well as estimated parameter values were less accurate than both our MC and PA approaches even when we use a large number of nodes. We believe this is due to the high-frequency content of the filters, which cannot be captured well by standard quadrature schemes.
>
> Nevertheless, to address this concern directly in the manuscript, we will expand our related work section to clarify that nearly all GLM implementations are in discrete time. Regarding [17], although a full reproduction is not feasible, we will include a brief comparison in the supplementary material based on our partial reimplementation on a small simulated dataset, highlighting both the computational challenges and performance limitations we observed. This will help clarify the practical barriers to applying existing quadrature-based methods. We will also note that developing more scalable quadrature methods— those better suited for GPU acceleration or for estimating high-frequency coupling filters—remains a direction for future work.
>
> ### **Reviewer Specific Concerns**
>
> > The polynomial approximation methods (both PA-d and PA-c) are inherently less accurate compared to Monte Carlo (MC) or hybrid approaches.
>
> The reviewer is correct that the PA method is inherently less accurate due to its global approximation of firing rates. However, it enables much faster inference (especially with the closed-form solution), making it a good initialization procedure in the hybrid PA-MC model. In contrast, the sampling-based (MC) approach, while more accurate overall, can potentially suffer from slow inference since the number of required samples M grows proportional to the recording length.
>
> > the paper identifies "exploring variance reduction techniques for Monte Carlo sampling of the CIF" as a key future direction
>
> We highlight variance reduction for MC as an important direction for future work to address the speed limitation described above. In fact, we are currently working on a control variate technique (e.g., Chapter 8 in [Owen2013]), which significantly reduces the variance of the gradient estimates and decreases the required number of samples, resulting in faster optimization.
>
> > The current work primarily focuses on [...]  using an exponential nonlinearity
>
> While we focus on the exponential nonlinearity in the main text, we also use softplus nonlinearity in the supplement. In general, both the MC and PA approaches are compatible with any nonlinearity that does not violate the convexity of the log-likelihood [1].
>
> > …and restricts its analysis to spike history.
>
> The input covariates X can be easily extended to include common external variables such as animal position or visual stimulus presentation. This would involve an additional term in the log-rates that include new parameters plus some observed experimental covariate like position or stimulus type. Here, we restrict our analyses to spike history since we are focused on these fast transient dynamics specially as a proxy to anatomical connectivity.
>
> > Maximum likelihood estimation in autoregressive GLMs is often prone to [...] self-excitation
>
> This phenomenon does indeed apply here. However, for the data simulation process, we ensure that the (autoregressive) self-filters are strictly negative to prevent this very self-excitation. We did not observe any self-excitation issues in the MLE estimates of post-spike filters, in simulated or real datasets.
>
> > How does the overall firing rate influence the performance of the method?
>
> The effect of overall firing rate in our model is the same as that observed in discrete-time GLMs. High-spiking neurons are generally easier to fit because they provide more data, and even small coupling weights are sufficient to noticeably influence the predicted firing rate. In contrast, low firing rate neurons require a large negative bias term to match their background firing rate. This suppresses the impact of inputs, meaning that only large coupling weights can produce meaningful post-synaptic effects. Overall, low-spiking neurons often exhibit lower signal-to-noise ratio and present a challenge for accurate inference of connectivity. This is an important point, and we have added a sentence to the manuscript to make this limitation clear to the reader.
>
> **Additional references**
>
> [Chen2009] Chen, Z., Putrino, D. F., Ba, D. E., Ghosh, S., Barbieri, R., & Brown, E. N. (2009). A regularized point process generalized linear model for assessing the functional connectivity in the cat motor cortex. Annual International Conference of the IEEE Engineering in Medicine and Biology Society. IEEE
>
> [Owen2013] Owen, A. B. (2013). Monte Carlo theory, methods and examples.

---

> > ### Comment · Reviewer_8hfS · 2025-08-03
> >
> > I thank the authors for addressing my questions and concerns in their rebuttal. I am mostly satisfied with the response, and will maintain my score.

---

> > > ### Author Response · Authors · 2025-08-05
> > >
> > > We thank the reviewer for the prompt acknowledgement and feedback. If there are any parts of the response that they felt insufficiently addressed their concerns, we are more than happy to clarify further. If the reviewer believes that the clarifications and changes to the manuscript strengthen the submission, we are hoping that they are open to modifying their score.

---

### Official Review · Reviewer_Adrb · 2025-06-25

**Clarity:** 2
**Significance:** 3
**Originality:** 3
**Rating:** 4
**Confidence:** 4

**Summary:**

The manuscript proposes two new algorithms to optimize continuous-time Poisson Generalized Linear Models (GLM) on spike train data. This allows to infer dynamics at submillisecond timescales and scale this to long spike trains without the memory overhead that is introduced by binning the data into spike counts. For this the authors approximate the cumulative intensity function (CIF) that is part of the continuous-time GLM either using Monte Carlo (MC) or polynomial approximations (PA) of the integral. Using Laguerre polynomials as a basis function for the latter even admits a closed form solution. The method is demonstrated both on real and synthetic data, showing the proposed algorithms scale better in the number of neurons and time-steps. Using their algorithm they find functional connections in the hippocampus that are in line with what we know experimentally.

**Questions:**

- Do the authors know why the error in Fig. 2 is increasing with more > 3 basis functions and is this something expected? If yes, this is not obvious to me. Could the authors elaborate on this? Would tuning using a Bayesian information criterion make sense?
- Something that is also not obvious to me is how the stratified sampling leads to lower variance compared to uniform sampling in L158? If the authors could explain this to me, that would be much appreciated.

**Ethical Concerns:**

["NO or VERY MINOR ethics concerns only"]

**Final Justification:**

The authors have addressed all my concerns during the rebuttal. I slightly err on the side of acceptance and have therefore raised my score from 3 to 4.

**Limitations:**

My feeling is that the authors insufficiently discuss potential shortcomings of their method.

**Paper Formatting Concerns:**

- Citation 30 is missing
- The formatting of Fig. S2 could be improved

**Quality:**

2

**Strengths And Weaknesses:**

## Strengths
- Using a continuous-time formulation of the GLM can take advantage of the sparse nature of spike trains. Instead of operating on discretized spike count arrays of size (1, num_time_bins), one can operatore on lists of spike times [t_1,t_2,...,t_N] whose size is independent of the discretization. While the idea to make these models tractable using polynomial approximations is not new, the authors propose to use a basis of Laguerre polynomials that permit a closed form solution and need fewer features to capture neural filters. Both of this improves computational efficiency, however accuracy does not seam to be effected (according to Fig. 4 and 5). Using a MC approximation is also a straight forward thing to try and gives the experimenter more options when deploying these methods. Both of these are small, but useful contributions in my view.
- The fact that the authors are able apply this method on hippocampal data and show method can recover known connectivity structures is also a nice demonstration of why I think this method could generally be useful.

## Weaknesses
- The paper overall feels a bit rough around the edges in its current state and it can be hard at times to follow the thought-process of the authors. While the method and its derivation are sound, the presentation could in my opinion be improved. I found that a lot of information is left for the reader to piece together (see comments and as follows). For example, I find the there could be a more natural flow between paragraphs of the experiments section that makes it easier to follow. In addition I had trouble understanding the exact content of Fig. 3, despite the associated paragraph and figure caption. What is the point it is making? Why is the error increasing after optimization in panel C? If the authors could explain this to me again / make this more clear, I would be appreciate this.
- Another issue I have with this method is that related work, while being mentioned, is barely discussed and no shortcomings are given.

## Comments
- L4 why not write Monte Carlo methods?
- On my printed out version of the manuscript it is hard to tell apart the similar shades of pink and red within the same plot (I.e. Fig. 1 B and D, Fig. 4,). It would be great if the authors could make them more differentiable.
- L142 The use of W here is a bit confusing, since it means something different from the W used elsewhere. Renaming this to $\tau$ or $\Delta T$ similar could make this less confusing.
- What is $I$ in Eq. 10? Was this explained?
- While I figured out what DB means in Fig. 3 from Fig 5.'s accompanying text it would be great to mention this somewhere earlier in the manuscript (or pot. the figure caption).
- In Fig. 4 you plot convergence time, but while you do mention what this means in App. 1.4., this is not linked
- The use and introduction of abbreviations (i.e. MC) is inconsistent and confusing. Sometimes abbrevs. are used, sometimes not, sometimes they are introduced multiple times.
- The supplement is sometimes referenced, but it would be nice if the authors could ref. the specific sections directly, i.e. in L273 or in L188.

---

> ### Author Rebuttal · Authors · 2025-07-31
>
> We thank the reviewer for their thoughtful questions, comments, and suggestions. We have made revisions in the manuscript to incorporate their feedback, and we elaborate on their criticisms and requests for clarification below:
>
> ### **Global Response**
>
> **Comparison to existing literature/baseline evaluation**
>
> All reviewers note that there is prior work on continuous GLMs and note a lack of comparison to these methods. However, much of the cited work are not actual model fits of a continuous GLM, and thus are not appropriate baselines for direct comparison. Some citations develop theoretical tools relevant to continuous Poisson point processes—such as convexity of the log-likelihood [1], or error bounds and information capacity [14]—and others use continuous-time models in latent variable settings [16]. Some of the other work mentioned, particularly [15] and [Chen2019] referenced by 8hfS are actually discrete-time implementations of GLMs, and so are captured in our discrete benchmark in figures 3, 4, and 5.
>
> To our knowledge, the only prior work that actually fits a continuous time Poisson GLM is [17] which uses Gauss-Lobatto quadrature to approximate the CIF. Unfortunately, the link provided in their manuscript to public code is no longer available. We attempted to implement their method but found that a naive implementation was both computationally infeasible on datasets of the size we consider and less accurate in approximating the CIF and fitting the model on very small subsets of data. Their approach relies on inserting a varying number of quadrature nodes between every spike time, which, for our hippocampal dataset with ~5 million spikes, would require storing and evaluating far more nodes than spikes—leading to significant memory issues and prohibitively slow inference (MC, by contrast, uses many times fewer samples than spikes). We have since tested their quadrature approach on dataset smaller than our current evaluations in the manuscript and found that the CIF estimate as well as estimated parameter values were less accurate than both our MC and PA approaches even when we use a large number of nodes. We believe this is due to the high-frequency content of the filters, which cannot be captured well by standard quadrature schemes.
>
> Nevertheless, to address this concern directly in the manuscript, we will expand our related work section to clarify that nearly all GLM implementations are in discrete time. Regarding [17], although a full reproduction is not feasible, we will include a brief comparison in the supplementary material based on our partial reimplementation on a small simulated dataset, highlighting both the computational challenges and performance limitations we observed. This will help clarify the practical barriers to applying existing quadrature-based methods. We will also note that developing more scalable quadrature methods— those better suited for GPU acceleration or for estimating high-frequency coupling filters—remains a direction for future work.
>
> **Limitations of our approach**
>
>
> We agree with the reviewers that the limitations of our approach were not sufficiently discussed, and we have expanded the conclusion section of the manuscript to address this. Many limitations of our model are inherited from the broader class of Poisson GLMs. This includes the challenge of dissociating monosynaptic connections from correlated firing (e.g. [6]) and the difficulty of identifying true connectivity without overly penalizing weak dependencies and connections with low spiking neurons ([24], [31], see response to reviewer 8hfS). Addressing these limitations is a priority for future work. Another limitation is the choice of the Poisson distribution itself, which may be a suboptimal match for describing spiking activity due to its assumptions about variance. Flexible alternatives such as the negative binomial distribution can potentially better capture the characteristics of neural spiking (e.g. [19] and [Pillow&Scott2012]).
>
> In response to z5Zq’s and 8hfS’s comments about individual contributions and shortcomings of our MC and PA approaches, we note that there is a clear trade-off between speed and performance. As 8hfS noted, the PA method is inherently less accurate due to its global approximation of firing rates. However, it enables much faster inference (especially with the closed-form solution). The sampling-based (MC) approach, while more accurate overall, can potentially suffer from slow inference since the number of required samples M grows proportional to the recording length. This limitation is partially overcome in the hybrid model through PA-based initialization.
>
> **Notation**
>
> We thank the reviewers for drawing our attention to the notation errors. We have made the necessary revisions to ensure that our use of notation and abbreviations is introduced clearly and consistently in the correct order. This includes consistent use of $\textbf{w}$ and $x_t$ in Eq. 1, 2 and throughout the text; a more thorough introduction of the notation used in Equation 4; and consistent use of abbreviations (e.g., MC, GLM, DB). We have also aligned the notation used in the PA-c description with that of the MC method.
>
> ### **Reviewer Specific concerns**
> > I find there could be a more natural flow between paragraphs of the experiments section
>
> We have revised the manuscript to improve the overall flow of the experiments section. Specifically, we added a heading to the subsection with the analysis presented in Fig.3 (Stochastic gradient variance in discrete and continuous GLM)  and added clearer transitions at the beginning of the simulated and real data experiments.
>
> >I had trouble understanding the exact content of Fig. 3
>
> We apologize for the confusion here. The key point of this section is that the variability of the gradient estimates in the discrete GLM is exceptionally high when we are required to update the model in batches. Thus, gradient-based optimization converges very slowly. In contrast, using stratified MC samples yields much more stable inference with much better log-likelihood values. This is because for our MC approach, the first term in the log-likelihood is computed exactly and the CIF integral is sampled over the entire recording. The DB method however performs updates on a small subset of data, resulting in high gradient inaccuracy, as shown in Fig.3 A and C..
>
> We believe this confusion may be caused by an error in the caption to Fig.3C. This panel does not show the variance of the gradient estimates, but rather the expected squared error between the true and stochastic gradients or, in other words, the variability of the stochastic gradients around the true values. This error increases toward the end of training, since accurately estimating increasingly small gradient steps becomes more difficult as the models approach convergence. We clarify this distinction in the supplemental material and have updated the manuscript text accordingly. We also added the equation for the squared gradient error displayed in panel C: $\mathbb{E}\frac{||\nabla_p-\tilde{\nabla}_p||^2_2}{||\nabla_1||^2_2}$ where $\nabla_p$ is the true gradient at parameter update p and $\tilde{\nabla}_p$ is the corresponding stochastic gradient.
>
> > Comments **(additional notation points)**
>
> We are grateful for the detailed feedback on notation. We have incorporated the necessary changes into the manuscript, including updating the notation for $W$ (now $H$) and $\mathbf{W}$ (now  $\mathbf{w}$), introducing $I$ before Eq.10, and clarifying the wording in line 4. We also added references to relevant sections of the supplementary material where appropriate and fixed citation [30]. We also adjusted the colors used for purple, red, and pink to improve their distinguishability in print, and we rearranged the panels in Figure S2 to a horizontal layout to eliminate excess white space.
>
> > why the error in Fig. 2 is increasing with more > 3 basis functions
>
> The increase in error observed for both GL and RC bases on the simulated dataset is due to overfitting and was seen specifically for these simulated filters (the introduction of ridge regularization changes this trend somewhat for these, but does not improve overall fits).
> However, this is not true for all datasets. We repeated the analysis from Fig. 2 on a subset of the real hippocampal dataset from Fig. 5 and found that, for both basis choices, the log-likelihood on held-out data improved as more basis functions were added. In this case, we use cross-validated log-likelihood as an evaluation metric, though BIC would also be a valid choice. Moreover, the GL basis outperformed the RC basis across all set sizes. We will include these results as an additional panel in Fig. 5 and discuss these points in the manuscript.
>
> > how the stratified sampling leads to lower variance compared to uniform sampling
>
> The intuition behind stratified sampling is that it ensures a more uniform coverage of the CIF domain, avoiding the clustering and gaps that can occur with purely uniform samples. For example, if we drew M=10 samples, we might get unlucky and all 10 samples would clump together at a similar time. Intuitively, we do better by dividing the recording into 10 equally sized time intervals and drawing one sample per interval. This forces the samples to be more spread out. For more details, see Chapter 8 in [Owen2013].
>
> **Additional references**
>
> [Chen2009] Chen, Z., Putrino, D. F., Ba, D. E., Ghosh, S., Barbieri, R., & Brown, E. N. (2009). A regularized point process generalized linear model for assessing the functional connectivity in the cat motor cortex.  IEEE
>
> [Owen2013] Owen, A. B. (2013). Monte Carlo theory, methods and examples.
>
> [Pillow&Scott2012] Pillow, J., & Scott, J. (2012). Fully Bayesian inference for neural models with negative-binomial spiking. NIPS 2025.

---

> > ### Comment · Reviewer_Adrb · 2025-08-04
> > **reply to author rebuttal**
> >
> > Thank you to the authors for the clarifications, additions to the manuscript and fixing the formatting of equations, figures and captions. While I think the overall presentation could be improved further, all my immediate concerns have been addressed by the authors. I will therefore raise my score.

---

### Official Review · Reviewer_z5Zq · 2025-06-29

**Clarity:** 2
**Significance:** 2
**Originality:** 1
**Rating:** 4
**Confidence:** 3

**Summary:**

This paper used continuous time Poisson process GLM to infer functional connectivity and proposed two methods to approximate the cumulative intensity function in the log likelihood equation: Monte Carlo stratified sampling, and polynomial approximation. They also used Laguerre polynomials as basis functions for GLM temporal filters. Empirical results show that these methods achieve better convergence with less runtime than discrete batched GLM. The methods can uncover some features of hippocampus connectivity after pre-selecting temporal filters within a certain timescale.

**Questions:**

1. What is the main difference between this paper and previous continuous time Poisson process GLM?

2. Do methods in this paper outperform those in [1] and [15] when used to infer functional connectivity?

3. If so, what is the individual contribution of Monte Carlo stratified sampling, polynomial approximation, choice of temporal filters and choice of nonlinearities?

4. The authors pre-selected filters within a certain time scale (line 297) when using the model on a hippocampal dataset. In general, how much prior is required when using this method on a real dataset?

**Ethical Concerns:**

["NO or VERY MINOR ethics concerns only"]

**Final Justification:**

The authors have addressed my my concerns during the rebuttal and I have increase my score as a result.

**Limitations:**

See above.

**Quality:**

3

**Strengths And Weaknesses:**

The framework of this paper is clear, and the methods proposed in this paper achieve better performance than the discrete time batched GLM. However, continuous time Poisson process GLM is not a new concept and has been studied extensively in the field, as is pointed out in the introduction section with citations [1][14][15][16][17], and this paper has no direct comparison to these previous models or methods.
Additionally, there are several places that should be written more carefully:
Equation (1) and (2) summarize methods for discrete time Poisson GLM, but the notations are not treated carefully: $X_t$ is never introduced, and I think the authors meant to write $x_t$ in (1), but then $W^T X_t^T$ will be confusing in (2). These equations directly correspond to equations (1) and (7) of the cited paper [8], which have very clear notations, so maybe the authors should just follow that.
Figure 3A and B has no y-axis values, making it hard to assess whether batched discrete time GLM ‘converges poorly’ as stated in line 233. Line 237 references Figure 3C instead of Figure 3B as stated. Line 245 references Figure 3B instead of Figure 3A as stated.
The hybrid PA-MC model introduced in line 254 should be explained in more detail, especially given that it is the best performing method.

---

> ### Author Rebuttal · Authors · 2025-07-31
>
> We thank the reviewer for their comprehensive review and detailed feedback. In response to the reviewer’s main criticism regarding insufficient comparison to prior work, we provide the following general reply that addresses this issue across the reviewer’s comments.
>
> ### **Global Response**
>
> **Comparison to existing literature/baseline evaluation**
> All reviewers note that there is prior work on continuous GLMs and note a lack of comparison to these methods. However, much of the cited work are not actual model fits of a continuous GLM, and thus are not appropriate baselines for direct comparison. Some citations develop theoretical tools relevant to continuous Poisson point processes—such as convexity of the log-likelihood [1], or error bounds and information capacity [14]—and others use continuous-time models in latent variable settings [16]. Some of the other work mentioned, particularly [15] and [Chen2019] referenced by 8hfS are actually discrete-time implementations of GLMs, and so are captured in our discrete benchmark in figures 3, 4, and 5.
>
> To our knowledge, the only prior work that actually fits a continuous time Poisson GLM is [17] which uses Gauss-Lobatto quadrature to approximate the CIF. Unfortunately, the link provided in their manuscript to public code is no longer available. We attempted to implement their method but found that a naive implementation was both computationally infeasible on datasets of the size we consider and less accurate in approximating the CIF and fitting the model on very small subsets of data. Their approach relies on inserting a varying number of quadrature nodes between every spike time, which, for our hippocampal dataset with ~5 million spikes, would require storing and evaluating far more nodes than spikes—leading to significant memory issues and prohibitively slow inference (MC, by contrast, uses many times fewer samples than spikes). We have since tested their quadrature approach on dataset smaller than our current evaluations in the manuscript and found that the CIF estimate as well as estimated parameter values were less accurate than both our MC and PA approaches even when we use a large number of nodes. We believe this is due to the high-frequency content of the filters, which cannot be captured well by standard quadrature schemes.
>
> Nevertheless, to address this concern directly in the manuscript, we will expand our related work section to clarify that nearly all GLM implementations are in discrete time. Regarding [17], although a full reproduction is not feasible, we will include a brief comparison in the supplementary material based on our partial reimplementation on a small simulated dataset, highlighting both the computational challenges and performance limitations we observed. This will help clarify the practical barriers to applying existing quadrature-based methods. We will also note that developing more scalable quadrature methods— those better suited for GPU acceleration or for estimating high-frequency coupling filters—remains a direction for future work.
>
> **Limitations of our approach**
>
> We agree with the reviewers that the limitations of our approach were not sufficiently discussed, and we have expanded the conclusion section of the manuscript to address this. Many limitations of our model are inherited from the broader class of Poisson GLMs. This includes the challenge of dissociating monosynaptic connections from correlated firing (e.g. [6]) and the difficulty of identifying true connectivity without overly penalizing weak dependencies and connections with low spiking neurons ([24], [31], see response to reviewer 8hfS). Addressing these limitations is a priority for future work. Another limitation is the choice of the Poisson distribution itself, which may be a suboptimal match for describing spiking activity due to its assumptions about variance. Flexible alternatives such as the negative binomial distribution can potentially better capture the characteristics of neural spiking (e.g. [19] and [Pillow&Scott2012]).
>
> In response to z5Zq’s and 8hfS’s comments about individual contributions and shortcomings of our MC and PA approaches, we note that there is a clear trade-off between speed and performance. As 8hfS noted, the PA method is inherently less accurate due to its global approximation of firing rates. However, it enables much faster inference (especially with the closed-form solution). The sampling-based (MC) approach, while more accurate overall, can potentially suffer from slow inference since the number of required samples M grows proportional to the recording length. This limitation is partially overcome in the hybrid model through PA-based initialization.
>
> **Notation**
>
> We thank the reviewers for drawing our attention to the notation errors. We have made the necessary revisions to ensure that our use of notation and abbreviations is introduced clearly and consistently in the correct order. This includes consistent use of $\textbf{w}$ and $x_t$ in Eq. 1, 2 and throughout the text; a more thorough introduction of the notation used in Equation 4; and consistent use of abbreviations (e.g., MC, GLM, DB). We have also aligned the notation used in the PA-c description with that of the MC method.
>
> ### **Reviewer specific concerns**
>
> > Figure 3A and B has no y-axis values,  Line 237 and Line 245 references
>
> We apologize for the confusion with Fig.3. We have corrected the references to panels in B (line 245) and C (line 237) as pointed out by the reviewer. We additionally have adjusted the plots to include the log-likelihood values. However, want to emphasize the the purpose of this plot is to demonstrate that the discrete GLM batching approach yields highly variable log-likelihoods that remain far from the optimal log-likelihood even after many gradient steps (dark and light green lines in panel A), but a qualitatively similar stochastic approximation using the MC does not have this same issue (panel B). We have adjusted the wording in the manuscript to clarify this distinction. We also have clarified panel C adding the equation for squared gradient error (see reviewer Adrb response).
>
> > The hybrid PA-MC model [...] should be explained in more detail
>
> In this approach, we first fit a PA-c model and then use its parameter estimates to initialize the stochastic gradient descent algorithm that leverages  MC. This initialization is closer to the true optimum and thus helps to reduce the number of optimization steps needed and speeds up inference (a.k.a. a “warm start”). We have revised the manuscript to clarify this.
>
> > what is the individual contribution of [...] choice of temporal filters
>
> We found that our proposed Laguerre basis outperforms the standard raised cosine basis consistently across datasets in a variety of settings and hyperparameter values. In addition to the comparison on simulated data (Fig. 2), we have expanded results on the hippocampal dataset (Fig.5), where the Generalized Laguerre basis achieves higher cross-validated log-likelihoods across a range of basis set sizes (2 to 7). In both cases for these data, performance improves as more basis functions are added. We also added a panel to Fig.S2 in the supplement illustrating the effect of introducing ridge regularization to fits on simulated data.
>
> > and choice of nonlinearities?
>
> Although the exponential inverse link provides a clear benefit in terms of closed-form solution for the PA-c model, we find that the softplus nonlinearity generally results in more stable inference for both PA-c and MC. Additional evaluations using softplus can be found in the supplement.
>
> >  In general, how much prior is required when using this method on a real dataset?
>
> We set the window for filter selection that reflects realistic synaptic dynamics, a transient synaptic delay followed by a fast interaction of 1-2 ms. While these properties are common across systems, we recommend using prior literature describing timescales of the interactions to guide this choice or running preliminary analyses with longer history windows before refining.
>
> > The paper referenced supplement material several times, but I do not see any appendix.
>
> The supplementary material is available as a downloadable .zip file included with the submission. The folder contains a PDF supplement, along with GPU-optimized model code and an example script. Please let us know if any part of the material is inaccessible.
>
> **Additional references**
>
> [Chen2009] Chen, Z., Putrino, D. F., Ba, D. E., Ghosh, S., Barbieri, R., & Brown, E. N. (2009). A regularized point process generalized linear model for assessing the functional connectivity in the cat motor cortex. IEEE
>
> [Owen2013] Owen, A. B. (2013). Monte Carlo theory, methods and examples.
>
> [Pillow&Scott2012] Pillow, J., & Scott, J. (2012). Fully Bayesian inference for neural models with negative-binomial spiking. NIPS 2025

---

> > ### Comment · Reviewer_z5Zq · 2025-08-04
> >
> > Thank you for the detailed response and clarifications. Including these clarifications in the manuscript, especially about comparisons with related work, will further strengthen the paper. Also good to know that the proposed Laguerre basis outperforms raised cosine basis on the real dataset as well, and that discussions about softplus nonlinearity was included in the supplement. I will raise my overall rating to 4.

---

### Official Review · Reviewer_D8KZ · 2025-07-03

**Clarity:** 3
**Significance:** 3
**Originality:** 3
**Rating:** 5
**Confidence:** 4

**Summary:**

The authors address an important issue in the analysis of spiking neural data: how to reasonably apply Poisson GLM without discretizing into binned count data in order to estimate functional connectivity at the temporal scale of synaptic events. Current binning approaches degrade temporal resolution and for modern data sets leads to computational limitations. The authors continuous formulation uses two tractable surrogates for the intractable cumulative intensity integral: stratified Monte-Carlo quadrature (MC) and a quadratic polynomial approximation (PA-c) that gives a closed form MAP update. They further introduce an exponentially scaled Laguerre basis as an orthogonal alternative to the raised cosine basis.They validate their methods with one Neuropixels data set, showing that MC/PA-c run faster than batched discrete GLMS, recover filters more accurately than discrete or PA-d methods, and provide better estimates of hippocampal connectivity. They provide GPU-accelerated code.

**Questions:**

How novel is the Laguerre basis? Is this not used in earlier work?

Why not use public large-scale data sets such as Steinmetz et al. visual cortex data and baseline against previous continuous Hawkes process models and quadrature methods from Paninksi, Lindermann, and others?

Does MC becomes prohibitively slow for high firing rates or when PA-c becomes memory-bound (long recordings, hundreds or thousands of neurons)?

**Ethical Concerns:**

["NO or VERY MINOR ethics concerns only"]

**Final Justification:**

We are satisfied with the responses provided by the reviewer, provided they include the full analysis (N=642) they suggest. We therefore raise our score to a 5.

**Limitations:**

The limitations could be better highlighted in the main paper.

**Quality:**

3

**Strengths And Weaknesses:**

**Strengths**

This problem is of increasing importance as neural recording methods improve in speed and scale.
The quadratic PA-c derivation is elegant
The Laguerre function drop preserves biophysical realism
Thorough wall-time and memory comparisons
GPU-optimized code provided

**Weaknesses**

The paper would benefit from more comprehensive baselining. For example, it currently evaluates only against a binned Poisson-GLM variant (PA-d/SVRG) and one small Neuropixels data set.
How do things scale as the number of neurons increases?

**Minor**

Notation introduction could be made clearer: for example, not introducing what x_s is at line 140 well after it’s used in Eq 4. This is not the only case of late introduction of variables.

---

> ### Author Rebuttal · Authors · 2025-07-31
>
> We thank the reviewer for their detailed and insightful feedback. Below are our responses to the reviewer’s comments and questions.
>
> ### **Global Response**
>
> **Comparison to existing literature/baseline evaluation**
>
> All reviewers note that there is prior work on continuous GLMs and note a lack of comparison to these methods. However, much of the cited work are not actual model fits of a continuous GLM, and thus are not appropriate baselines for direct comparison. Some citations develop theoretical tools relevant to continuous Poisson point processes—such as convexity of the log-likelihood [1], or error bounds and information capacity [14]—and others use continuous-time models in latent variable settings [16]. Some of the other work mentioned, particularly [15] and [Chen2019] referenced by 8hfS are actually discrete-time implementations of GLMs, and so are captured in our discrete benchmark in figures 3, 4, and 5.
>
> To our knowledge, the only prior work that actually fits a continuous time Poisson GLM is [17] which uses Gauss-Lobatto quadrature to approximate the CIF. Unfortunately, the link provided in their manuscript to public code is no longer available. We attempted to implement their method but found that a naive implementation was both computationally infeasible on datasets of the size we consider and less accurate in approximating the CIF and fitting the model on very small subsets of data. Their approach relies on inserting a varying number of quadrature nodes between every spike time, which, for our hippocampal dataset with ~5 million spikes, would require storing and evaluating far more nodes than spikes—leading to significant memory issues and prohibitively slow inference (MC, by contrast, uses many times fewer samples than spikes). We have since tested their quadrature approach on a dataset smaller than our current evaluations in the manuscript and found that the CIF estimate as well as estimated parameter values were less accurate than both our MC and PA approaches even when we use a large number of nodes. We believe this is due to the high-frequency content of the filters, which cannot be captured well by standard quadrature schemes.
>
> Nevertheless, to address this concern directly in the manuscript, we will expand our related work section to clarify that nearly all GLM implementations are in discrete time. Regarding [17], although a full reproduction is not feasible, we will include a brief comparison in the supplementary material based on our partial reimplementation on a small simulated dataset, highlighting both the computational challenges and performance limitations we observed. This will help clarify the practical barriers to applying existing quadrature-based methods. We will also note that developing more scalable quadrature methods—those better suited for GPU acceleration or for estimating high-frequency coupling filters—remains a direction for future work.
>
> **Limitations of our approach**
>
> We agree with the reviewers that the limitations of our approach were not sufficiently discussed, and we have expanded the conclusion section of the manuscript to address this. Many limitations of our model are inherited from the broader class of Poisson GLMs. This includes the challenge of dissociating monosynaptic connections from correlated firing (e.g. [6]) and the difficulty of identifying true connectivity without overly penalizing weak dependencies and connections with low spiking neurons ([24], [31], see response to reviewer 8hfS). Addressing these limitations is a priority for future work. Another limitation is the choice of the Poisson distribution itself, which may be a suboptimal match for describing spiking activity due to its assumptions about variance. Flexible alternatives such as the negative binomial distribution can potentially better capture the characteristics of neural spiking (e.g. [19] and [Pillow&Scott2012]).
>
> In response to z5Zq’s and 8hfS’s comments about individual contributions and shortcomings of our MC and PA approaches, we note that there is a clear trade-off between speed and performance. As 8hfS noted, the PA method is inherently less accurate due to its global approximation of firing rates. However, it enables much faster inference (especially with the closed-form solution). The sampling-based (MC) approach, while more accurate overall, requires multiple iterations to converge to a solution. The hybrid model, which uses PA-based initialization followed by MC finetuning, is our attempt to enjoy the best of both approaches.
>
> **Notation**
>
> We thank the reviewers for drawing our attention to the notation errors. We have made the necessary revisions to ensure that our use of notation and abbreviations is introduced clearly and consistently in the correct order. This includes consistent use of $\textbf{w}$ and $x_t$ in Eq. 1, 2 and throughout the text; a more thorough introduction of the notation used in Equation 4; and consistent use of abbreviations (e.g., MC, GLM, DB). We have also aligned the notation used in the PA-c description with that of the MC method.
>
> ### **Reviewer Specific concerns**
>
> >How novel is the Laguerre basis? Is this not used in earlier work?
>
> Laguerre polynomials are typically found in the physics literature (e.g. [Quesne2009]) but have been used as bases for temporal fMRI signals ([Solo2004]). To the best of our knowledge, our paper is the first one to adopt them as basis functions for constructing neuronal filters.
>
> >baseline against previous continuous Hawkes process models and quadrature methods from Paninksi, Lindermann, and others
>
> See the general response above. To the best of our knowledge, our work presents the first implementation of a fully continuous-time GLM scalable to the dataset sizes analyzed in this paper, and we now include a comparison to quadrature on a small dataset in the supplement.
>
>  We also thank the reviewer for pointing out Hawkes processes—a relevant line of work that we had overlooked. We have expanded the review of related work in the manuscript to include a discussion of the distinctions between Hawkes models and our Poisson point process GLM framework. Importantly, Hawkes processes only allow for excitatory interactions between neurons and cannot account for inhibitory interactions. On the other hand, the nonnegativity assumption built into a Hawkes process facilitates certain tricks (in particular, leveraging the superposition property of Poisson point processes). At the end of the day, both models have their virtues. Our work focuses on finding better optimization algorithms for one of them (the point process GLM), and it doesn’t make sense to benchmark optimization methods across the two since the models have fundamentally different structure. [Linderman&Adams].
>
> >Does MC become prohibitively slow for high firing rates or when PA-c becomes memory-bound (long recordings, hundreds or thousands of neurons)?
>
> Our continuous-time GLM framework works with a compact data representation (marked spike-time sequences) whose storage requirements do not scale with the number of neurons or the recording duration directly. Instead, the bottleneck is the total number of spikes in the recording which becomes limiting at the scale of tens of millions of spikes.
>
> The reviewer is correct that the MC and PA-c methods impose distinct computational constraints. MC-based optimization requires iterating over all spike times and MC samples at each gradient step. However, the number of MC samples is independent of the number of neurons (unlike quadrature-based methods discussed above) and scales only with the recording length. In our analyses of large-scale datasets, we find that the number of MC samples remains well below the number of spikes. In contrast, PA-c approach precomputes sufficient statistics in a single pass of the data. However, this step can lead to memory overhead due to the need to store large intermediate arrays, particularly the matrix of pairwise interactions M, which grows quadratically with the number of neurons. This limits parallelization and may slow down inference in larger (>1000) neural populations.
>
> >The paper […] currently evaluates only one small Neuropixels data set. How do things scale as the number of neurons increases?
>
> In the main text, we restricted our analysis of hippocampal data to a single probe to eliminate probe-specific artifacts that led to inferring artificial filters and confound connectivity estimation. However, we also ran preliminary experiments on the full recording (N=642 neurons) using both the MC and PA-c models. On this scale, the inference was slower (under an hour for MC and under 30 minutes for PA-c) but models fit without memory issues. If the reviewer believes these results would strengthen the submission, we can include them in the supplementary material along with a discussion of their limitations due to probe artifacts.
>
>
> **Additional References**
>
> [Chen2009] Chen, Z., Putrino, D. F., Ba, D. E., Ghosh, S., Barbieri, R., & Brown, E. N. (2009). A regularized point process generalized linear model for assessing the functional connectivity in the cat motor cortex. IEEE
>
> [Linderman&Adams] Linderman, S. & Adams, R. (2015). Scalable Bayesian Inference for Excitatory Point Process Networks. arXiv:1507.03228.
>
> [Owen2013] Owen, A. B. (2013). Monte Carlo theory, methods and examples.
>
> [Pillow&Scott2012] Pillow, J., & Scott, J. (2012). Fully Bayesian inference for neural models with negative-binomial spiking. Advances in neural information processing systems, 25.
>
> [Solo2004] V. Solo, C. J. Long, E. N. Brown, E. Aminoff, M. Bar and S. Saha. (2004). fMRI signal modeling using Laguerre polynomials. IEEE
>
> [Quesne2009] Quesne, C. (2009). Solvable Rational Potentials and Exceptional Orthogonal Polynomials in Supersymmetric Quantum Mechanics. SIGMA 5, 084.

---

> > ### Comment · Reviewer_D8KZ · 2025-08-04
> > **Score increase to a 5**
> >
> > We thank the authors for their thoughtful response. We do believe that they should include the full recording (N=642) neurons (and the related and interesting caveats related to confounds and filter inference), as these are quite relevant to real-world applications of their method. Assuming this is included in the final draft, we are satisfied and raise our score to a 5.

---

### Note · Authors · 2025-08-11

We thank the reviewers again for their thoughtful and constructive feedback. We are pleased and encouraged by the reviewers’ positive evaluation of our work. Alongside the changes we implemented during the initial response, we have added the full Neuropixels dataset analysis to the supplement, as per D8KZ’s suggestion.

We are pleased that 3 out of 4 reviewers have agreed to raise their scores, with all ratings now above threshold. While reviewer 8hfS maintained their score, they indicated they were mostly satisfied; we invited them to request any further clarifications but did not receive additional questions.

---

### Decision · Program_Chairs · 2025-09-17

**Decision:**

Accept (poster)

**Comment:**

This paper presents new methods for fitting continuous-time Poisson GLMs to spike train data, avoiding the limitations of binning. The authors propose two approaches—a Monte Carlo estimator and a polynomial approximation with Laguerre polynomials—that together make it possible to recover functional connectivity at submillisecond resolution while scaling to large recordings. The approach is validated on both synthetic data and hippocampal Neuropixels recordings, with open-source GPU implementations provided.

The main strengths are the clear computational benefits of the continuous-time formulation, the elegant closed-form update enabled by the Laguerre basis, and the demonstration on real data. These are practical contributions that address a timely need as datasets continue to grow in size and precision. Reviewers highlighted that the work is technically solid, the methods perform better than discrete baselines, and the release of optimized code increases the potential impact.

The initial submission had weaknesses in presentation and related work, with limited discussion of comparisons to earlier continuous-time models and some notational issues. The rebuttal addressed these points: the authors clarified prior work, added supplementary comparisons, expanded the discussion of limitations, and included results on the full dataset (N=642 neurons). After discussion, most reviewers raised their scores, with consensus that this is a valuable contribution. I recommend acceptance.